# Multiple Drug Resistant *Streptococcus* Strains—An Actual Problem in Pig Farms in Western Romania

**DOI:** 10.3390/antibiotics13030277

**Published:** 2024-03-19

**Authors:** Luminita Costinar, Corina Badea, Adela Marcu, Corina Pascu, Viorel Herman

**Affiliations:** 1Department of Infectious Diseases and Preventive Medicine, Faculty of Veterinary Medicine, University of Life Sciences “King Mihai I”, 300645 Timisoara, Romania; luminita.costinar@usvt.ro (L.C.); corina.badea@usvt.ro (C.B.); viorel.herman@fmvt.ro (V.H.); 2Department of Animal Production Engineering, Faculty of Bioengineering of Animal Recourses, University of Life Science “King Mihai I”, 300645 Timișoara, Romania; adelamarcu@usvt.ro

**Keywords:** *Streptococcus*, multidrug resistance, pig

## Abstract

Streptococci are a type of bacteria that can cause severe illnesses in humans and animals. Some typical species like *S. suis*, or atypical species like *S. porcinus* and, *S. dysgalactiae subsp. dysgalactiae*, can cause infections like septicemia, meningitis, endocarditis, arthritis, and septic shock. *S. suis* is considered a newly emerging zoonotic pathogen. Although human streptococcal infection outbreaks are rare, it is appropriate to review the main streptococcal species isolated in pig farms in western Romania, due to the high degree of antibiotic resistance among most isolates commonly used in human treatment. This study examines the resistance patterns of these isolates over 5 years (2018–2023). The research investigated the antimicrobial susceptibility of 267 strains of *Streptococcus* spp. isolated from pigs, primarily from lung and brain tissues. This report is the first to describe the distribution of atypical *Streptococcus* species (SDSE, *S. porcinus*, *S. hyovaginalis*, *S. pluranimalium*, *S. canis*) in Romania, as well as the antibiotic resistance profile of these potentially zoonotic species. It is important to re-evaluate and consider the high rates of resistance of *S. suis* to tetracyclines, lincosamides, macrolides, and aminoglycosides, as well as the high recovery rates of *S. suis* from the lungs and brain when treating swine diseases.

## 1. Introduction

Streptococci can cause a range of diseases in both humans and animals, from mild to severe. Normally, these bacteria belong to the commensal microflora but can sometimes cause infections as opportunistic pathogens. In 2020, several zoonotic streptococci were identified, including S. *suis*, *S. canis*, *S. dysgalactiae subsp. dysgalactiae*, *S. equi subsp. zooepidemicus*, *S. halichoeri*, *S. iniae*, *S. porcinus*, and others. Under certain circumstances, these and other streptococcal species can also cause human diseases [1].

Although *S. suis* is the most important species for the global pig industry, other *Streptococcus* species may also pose a greater and often underestimated risk to pig health by causing diseases. *Streptococcus suis* possesses antigens somehow related to Lancefield group D streptococcus. It is considered a facultatively anaerobic, alpha-hemolytic, Gram-positive, nonmotile coccus, displayed in chains of varying lengths. The number of reported human cases due to *S. suis* has increased significantly in recent years, mainly in China and Southeast Asian countries, especially in Vietnam and Thailand [2].

*S. dysgalactiae subsp. equisimilis* (*SDSE*) is a type of bacteria that belongs to the group of beta-hemolytic streptococci. It is typically found in the vaginal secretions and colostrum of sows and is considered to be a part of their commensal microflora. It is a common source of infection in piglets. *SDSE* can cause septicemia by entering the bloodstream through skin lesions or the tonsils of piglets and can lead to arthritis, meningitis, or endocarditis [3,4,5].

*Streptococcus porcinus*, a beta-hemolytic streptococci in Lancefield group NG1 (A1, C1), NG2, NG3, E, P, U, or V antigen, was first identified in pigs in 1984 and later detected in bovine milk. This species of streptococci is commonly associated with abortion, endocarditis, and pyogenic infections in pigs. It can colonize the genital and upper respiratory tracts of pigs, sheep, rabbits, dogs, guinea pigs, and cattle, making them a potential reservoir for the bacteria. Although a few cases of *S. porcinus* infections have been reported in humans, mostly as genitourinary tract infections in women, there is no clear data on the habitat and virulence properties of other streptococcal species such as *S. pluranimalium*, *S. parcorum*, *S. gallolyticus gallolyticus*, and *S. plurextorum*, which have also been isolated from sick pigs [6,7,8].

*Streptococcus pluranimalium* is a new species of streptococci found in various animal hosts. *S. pluranimalium* is associated with subclinical mastitis in dairy cows, reproductive disorders in cattle, valvular endocarditis, and septicemia in birds. It may also cause human infective endocarditis and brain abscesses. However, its pathogenicity mechanisms and virulence factors are still poorly understood. In Romania, this bacterium has not yet been isolated from animals [9,10]. It was first described in 1999 by Devriese et al. [11].

*Streptococcus canis* is a beta-hemolytic streptococcus and belongs to the G Lancefield group that colonizes the skin, upper respiratory tracts, ears, and reproductive tracts of dogs and cats [11,12]. *S. canis* is also an important pathogen causing skin infections in these species, as well as genitourinary tract infections, otitis externa, septicemia, pneumonia, endocarditis, septic arthritis, necrotizing fasciitis, etc. [11,12,13].

*S. canis* is found in various animals, causing bovine mastitis, and is also found in wild animals, such as mink, feral cats, and aquatic mammals [14]. It has been recognized as a zoonotic agent, with more studies reporting its isolation from skin diseases, soft tissue infections, bacteremia, and endocarditis in humans [13,15]. Since its discovery as a zoonotic agent in 1996, human cases of endocarditis, septicemia, cellulitis, and periprosthetic joint infections caused by *S. canis* have been reported [16,17,18].

This study aimed to provide an overview of the antimicrobial resistance patterns and the distribution of various species of *Streptococcus* spp. in different types of samples from clinically diseased pigs and from cadavers and to determine the potential pathogenic impact of other species of streptococci than *Streptococcus suis*.

## 2. Results

It can be observed from Table 1 and Figure 1 that the lungs (31.46%) are the primary source of streptococci, followed by the CNS-brain (21.72%), genitourinary tract (10.86%), liver, spleen (8.23%), serosal surfaces (7.49%), noses (6.36%), joints (4.86%), raw semen (3.37%), skin (2.99%) and others (2.62%).

Out of the 267 strains of streptococci that were isolated, *S. suis* was the most found, accounting for 67.79% of the samples, followed by *S. dysgalactiae subsp. equisimillis* (*SDSE*) at 14.98%, *S. porcinus* at 8.98%, *S. hyovaginalis* at 4.86%, *S. pluranimalium* at 1.87%, and *S. canis* at 1.49%. The identification of different species of streptococci was confirmed by matrix-assisted laser desorption ionization-time of flight (MALDI-TOF Biotyper System, Bruker Daltonics, Bremen, Germany).

Among diseased pigs exhibiting nervous manifestations, *S. hyovaginalis* was detected in 53.84%, SDSE in 27.50%, and *S. suis* in 22.09% of samples.

All streptococcus species, except *S. canis* and *S. pluranimalium*, were found in samples obtained from the lower respiratory tracts and particularly from the lungs. *S. suis* was isolated mostly from the lungs and brain, and, to a lesser extent, from the surface of the serosa.

During the study, it was found that *S. suis* was predominantly isolated from the lungs (37.01%) and brain (27.50%), and, to a lesser extent, from the surface of the serosa (9.39%). The samples from the genitourinary tract had the highest proportion of SDSE isolation (32.50%), followed by the brain (27.50%) and lungs (15.00%).

*S. porcinus*, on the other hand, was mostly isolated from the lungs (33.33%), followed by samples from the genitourinary tract (20.83%) and noses (16.66%).

Most strains *S. hyovaginalis* were isolated from the brain (53.84%), lungs (23.07%), and noses (15.38%).

Other *Streptococcus* spp. were isolated from various sources (heart and kidney), including the skin (33.33%), nose, organs, and raw semen (22.22% each). *S. suis* was the most isolated species.

### Antimicrobial Resistance Patterns

Out of the 26 antimicrobial substances tested, 13 strains of *S. suis* displayed resistance to 50% of them (Table 2). Only the data for isolated *S.* suis have been included in this table. The other streptococcus species isolated showed resistance below 50%. It is evident that there is a high frequency of resistance to some of the microbial agents, particularly in the *S. suis* strains isolated in the western part of Romania. Resistance is most frequent for lincomycin (95.00%), spectinomycin (94.44%), tylosin (87.50%), and doxycycline (87.23%), tilmicosin (85.71%), kanamycin (84.61%), chlortetracycline (83.67%), erythromycin (72.22%), apramycin (66.66%), trimethoprim–sulfamethoxazole (65.11%), and colistin (53.24%).

The antimicrobial resistance of zoonotic bacteria in swine-origin streptococci is concerning as it may compromise human infection treatment.

*SDSE* strains isolated in this study showed resistance to several antibiotics: ampicillin 22.50% (9/40), tylosin 32.50% (13/40), tilmicosin 30.00% (12/40), tulathromycin 25.00% (10/40), doxycycline 75.00% (30/40), chlortetracycline 80.00% (32/40), enrofloxacin 62.50% (25/40), erythromycin 47.50% (19/40).

*S. porcinus* has shown significant resistance to lincosamides (lincomycin) at a rate of 83.33% (20/24), macrolides (erythromycin) at a rate of 79.16% (19/24), and tetracyclines (doxycycline, chlortetracycline) at a rate of 70.83% (17/24) and 75.00% (18/24), respectively. The frequency of resistance for sulfonamides was 58.33% (14/24), and for streptomycin, it was 37.50% (9/24).

Similarly, *S. hyovaginalis* showed high resistance to tetracycline at a rate of 92.30% (12/13), chlortetracycline at a rate of 84.61% (11/13), and lincomycin at a rate of 53.84% (7/13).

*S. canis* and *S. plurianimalium* had remarkable resistance to amoxicillin–clavulanic acid at 88.88% (8/9), amoxicillin and tetracyclines at 77.77% (7/9), and penicillin at 55.55% (5/9).

After analyzing the data presented in Table 3, it was evident that several resistance patterns have emerged.

Among these isolates, twenty-two can be classified as multiresistant as they are resistant to three families of antibiotics. Seventy-four, ninety-one, and forty strains were resistant to, respectively, four, five, or six families of antibiotics. The most common patterns were found in the tetracyclines, macrolides, aminoglycosides, tetracyclines, and lincosamides groups.

## 3. Discussion

This retrospective study aimed to gather data on the isolation of streptococci from pig farms and expand the information about the antibiotic resistance of streptococcal species with potential zoonotic transmission. We lacked sufficient information about other diseases and previous antimicrobial treatments used in the pig farms from which the samples were collected. However, we believe that this study should be published because there is no data available on other streptococcal species found in pig farms in Romania with intensive rearing systems. The isolation of species such as *S. canis* highlights some unknown aspects, and therefore, we consider it important to bring it to attention.

*Streptococcus suis* is a significant concern in the pig industry and is particularly dangerous to piglets who have recently been weaned. It is an emerging zoonotic agent. It is important to note that *S. suis* can affect individuals near infected pigs, such as those who work in slaughterhouses or factories that process pork-meat products [19,20,21,22].

In a recent study conducted in Romania by Doma et al. (2021), it was found that the most common cause of pig mortality was *S. suis* (68–70%), followed by *E. coli* (30–31%) [23].

Although *S. suis* usually occurs more often in the upper respiratory tract than in the lower respiratory tract, *S. suis* has been detected more frequently in lung samples than in nasal swabs [20,24,25]. We noticed the same thing in our study; the highest percentage of strains (37.01%) was isolated from the lung and only 5.14% of the strains were isolated from the noses. *S. suis* was also detected quite frequently in the brain (20.57%) followed by serosal surfaces (9.71%), parenchymal organs (8.23%), joints (6.28%), the genitourinary tract (5.71%), raw semen (2.85%), and skin (1.14%).

All the strains of *S. suis* that we isolated in pigs were from the pigs with clinical signs of disease. These pigs were also positive for other pathogens like PRRS (Porcine Respiratory and Reproductive Syndrome), PCV 2 (Porcine Circovirus 2), *Mycoplasma hyopneumoniae*, and Influenza A. These infections could have resulted in bacterial overgrowth, which may have made it easier for *S. suis* to invade. In our study, *S. suis* was the only species that we found in all the sites we examined, which included the CNS, upper and lower respiratory tracts, skin, mucous membranes, joints, serosal surfaces, and raw semen.

After being considered non-pathogenic for many years, *SDSE* is now recognized as an important bacterial pathogen [26,27]. Recent epidemiological studies have shown an increasing number of invasive infections with *SDSE* in humans, often among immunocompromised patients and elderly patients with underlying co-morbidities, which suggests that this species is likely to become even more clinically important in the near future [28,29].

In pigs, *SDSE* can cause septicemia, which can spread to the CNS, lungs, joints, and genitourinary tract. Other researchers have previously described these locations [18,30,31,32].

A previous study Stanton et al. [33] showed that *SDSE* can produce aggressive skin lesions. In our study, we isolated *S. canis* and *S. pluranimalium*, in addition to *SDSE*, from pig skin.

It is particularly important to elucidate the possibility that *SDSE* crosses the interspecies barrier between pigs and humans through comprehensive epidemiological approaches [26,34]. Since the strains of streptococci isolated from pigs may be zoonotic pathogens, it is important to study their genotypes and antimicrobial resistance phenotypes to prepare for a potential public health hazard [35,36]. It has been found that *SDSE* is a significant pathogen in pigs that can cause various health issues such as neurological signs, reproductive and respiratory disorders, and skin lesions. These findings are consistent with those obtained by Oh et al. [4].

In 2020, a study conducted in Korea revealed that *SDSE* strains from pigs exhibited elevated minimum inhibitory concentration (MIC) values for macrolides, tetracyclines, and fluoroquinolones. These antibiotics are commonly prescribed to individuals infected with SDSE as a substitute for beta-lactams [4]. Consequently, further research with a more extensive sample of strains is necessary to ascertain the likelihood of *SDSE* transmission from animals to humans across various populations in Romania.

*S. porcinus* is the etiological agent of streptococcal lymphadenitis in growing pigs; it also causes throat abscesses, and has sometimes been isolated from pneumonia and sows’ endometritis [37]. However, the isolation of *S. porcinus* from the lower respiratory tract is still debatable as some authors consider it to be either a primary lung agent or a secondary lung agent [33,35]. In contrast to previous reports, we could not isolate *S. porcinus* from CNS, skin, liver, spleen, or joints [8,38,39].

In our research, *S. porcinus* was isolated from the lungs (33.33%), nasal cavities (16.66%), genitourinary tract (20.83%), serosa (12.50%), and raw semen (8.33%), but another study performed in Austria [30] reported isolation from the upper respiratory tract (2.34%) and lungs (6.25%) only.

In our study, *S. hyovaginalis* was isolated from the lungs of three specimens, from the noses of two specimens, and the CNS in seven specimens, but could not be isolated from genitourinary tract samples. Previous reports have shown that *S. hyovaginalis* can be isolated exclusively from sow genitourinary tract specimens but did not isolate bacteria from the upper and/or lower respiratory tracts [25].

*S. pluranimalium* is a species of streptococci whose involvement is less well known in the etiology of some human and animal infections [40]. Thus, some authors describe it as a new human pathogen [10,41,42,43] with major implications for human health, and therefore, we consider it particularly important to know more about the prevalence of this bacterium in pig herds in Romania, as this report is the first in Romania describing the isolation of *S. pluranimalium* from pigs.

*S. pluranimalium* and *S. canis* were not isolated from the CNS, genitourinary tract, lungs, and joints, but were recovered from nasal cavities, skin, raw semen, and parenchymatous organs, which may indicate that they are species that can colonize the skin and apparent mucous membranes and that their presence in raw semen and the liver and spleen may only be accidental contamination.

Originally, in 1986, *Streptococcus canis* was thought to be a canine and bovine pathogen exclusively but has since been isolated from a range of wild and domestic mammals: cats, rabbits, rats, foxes, mink, raccoons, seals, sea lions, otters, and badgers, as well as humans [25,34,44,45]. Along with other authors, we consider that *S. canis* may be capable of causing disease in several species, including pigs, making it one of the streptococcal pathogens with the widest host range [1,46,47,48,49]. The main hosts of this streptococcus species appear to be dogs and cats but we recovered from the nasal cavities, skin, and raw semen of pigs. Does the recovery of this species from pigs raise several questions about whether it is a species that can colonize skin and mucous membranes similar to what happens in dogs and cats? Or is it just accidental isolation?

In the context that our samples came from, pig farms with drastic biosecurity measures (no dogs on the farm), we can hypothesize that the transmission vector of *S. canis* could be humans.

Although in recent years the zoonotic potential of this bacteria has been widely accepted, scientific evidence remains limited [50,51]. In a retrospective study conducted at the University Hospital of Bordeaux between 1997 and 2002, *S. canis* was confirmed in 1% (*n* = 80/6404) of all *Streptococcus*-positive samples sent for culture [45]. Knowledge of the epidemiology of *S. canis* in both veterinary and human medicine is based on relatively few studies and information, and it is somewhat unclear how and to what extent transmission occurs between different animal species, as well as which risk factors predispose humans to *S. canis* infection [45,46].

It is also very important to note that more than one *Streptococcus* spp. isolate was recovered from animals with clinical signs in the pig herds from which the samples were taken. This may support the hypothesis that certain species not known to be primary disease agents are indeed an underestimated health risk for pigs and usually go undetected.

We could not demonstrate a causal association between clinical symptoms and the recovery of different *Streptococcus* spp.; further experiments and studies on a larger number of samples and information are needed for this purpose.

In Romania, antimicrobials are widely used in the animal husbandry sector, leading to widespread MDR in pig flocks; with this being very well highlighted both in this study and in other studies conducted by other researchers in our country. Thus, *S. suis* strains showed resistance to 50% (13/26) of the antimicrobial substances tested.

In 2022, the percentage of antimicrobial resistance to *S. pneumoniae* in humans in Romania was 48.3% according to the European Centre for Disease Prevention and Control [52].

High percentages of resistance were observed in our study for protein synthesis inhibitors, such as lincomycin (95.00%), chlortetracycline (83.67%), doxycycline (87.23%), erythromycin (72.22%), tilmicosin (85.71%), tylosin (87.50%), apramycin (66.66%), neomycin (48.83%), kanamycin (84.61%), streptomycin (84.21%), tulathromycin (45.45%), and flumequine (50.00%). These results are consistent with previous reports [23,38].

The results of our study are consistent with previous studies, suggesting *S. suis’* susceptibility to cell-wall-synthesis inhibitors, including beta-lactam antibiotics and penicillin (66.17%) [30,53,54,55,56].

The quality and safety of pork in these areas are questionable and it is believed to be the main source of human infection. Some authors suggest that exposure to pigs at work or consuming raw pork products increases the risk of *S. suis* infection [57,58,59,60,61].

The most common antimicrobials used in the treatment of *S. suis* infections and for which the percentages of susceptibility are still high are ceftiofur (87.91%), cefquinome (82.14%), amoxicillin-clavulanic acid (80.95%), amoxicillin (55.68%), ampicillin (72.72%), enrofloxacin (47.36%), florfenicol (53.48%), and penicillin (66.17%). For the treatment of *S. suis* infection, beta-lactamase (amoxicillin, amoxicillin/clavulanic acid, ceftiofur, cefquinome, ampicillin, and penicillin) and fluoroquinolones (enrofloxacin) are still the medicines of choice.

According to research conducted by O’Dea in Australia [62], a significant proportion of isolates were found to be resistant to both tetracycline (99.30%) and erythromycin (83.80%). The study also observed low levels of resistance to florfenicol (14.90%), penicillin G (8.10%), ampicillin (0.70%), and trimethoprim/sulfamethoxazole (0.70%). Interestingly, none of the isolates were found to be resistant to enrofloxacin. However, the global trend of increasing antimicrobial resistance among streptococcal species is becoming increasingly problematic.

In this study, all isolated strains were resistant to at least one class of antibiotics, and a fairly high percentage were resistant to three or more classes of drugs, which indicated substantial multidrug resistance (MDR).

Since 1980, there has been an increasing resistance of streptococci to antimicrobials commonly used in the pig industry, such as the tetracycline and macrolide group, around the world [50,60,63,64,65].

In Romania, there is currently no systematic collection of data or comparisons on the antimicrobial resistance of *S. suis* from pig farms to different antimicrobials. However, there are extensive global reports of the high resistance of these bacteria [24,51,64,65,66,67].

Our results also corroborate the results of Yongkiettrakul et al. (2019) [57] and Hernandez-Garcia et al. (2017) [68], who reported that the strains of streptococci isolated in their research became resistant to all classes of antibiotics used in pigs.

In the current context of reducing the use of antimicrobials in animals, the importance of different species of streptococci as zoonotic pathogens is increasing. Streptococci represent a threat to human health due to the lack of effective vaccines on the market; vaccines that could reduce the number of streptococci in pigs and therefore reduce the risk of human disease [69]. More and more frequently, the idea is emerging that, in addition to reducing the use of antibiotics on farms, biosecurity and improved health management must also be considered [70].

Therefore, these atypical *Streptococcus* spp. isolated from pigs may pose a greater underestimated health risk to pigs but also to humans and should be correctly identified by more complex and sensitive veterinary diagnostic methods.

## 4. Materials and Methods

### 4.1. Clinical Samples

This is a retrospective study based on data collected by the Faculty of Veterinary Medicine Timisoara. The data comprises 4783 samples. These samples were collected over five years, from January 2018 to January 2023. Samples were taken from pigs displaying various clinical symptoms such as respiratory problems, nervous disorders, skin lesions, arthritis, pneumonia, genitourinary tract infections, and septicemia. All samples were collected by veterinarians and sent to the Faculty of Veterinary Medicine Timisoara—Department of Infectious Diseases and Preventive Medicine for routine diagnostic purposes.

A total of 267 samples of *Streptococcus* spp. were obtained from pigs of all age categories (sows, gilts, fattening pigs, weaned pigs, and suckling piglets) out of 4783 samples examined (Table 4).

The samples came from 40 pig farms in the western and north-western parts of Romania, from the counties with the highest density of pigs in the country. The farms are intensive, closed systems, which are breeding, farrow-to-finish and fattening farms. Semen samples were taken from a semen center.

### 4.2. Bacterial Isolation and Identification

In this study, we describe the detection of *Streptococcus* (*S.*) *suis*, *S. dysgalactiae subsp. equisimilis* (*SDSE*), *S. hyovaginalis*, *S. porcinus*, and other streptococcus species (*S. canis*, *S. pluranimalium*) isolated from various types of samples. Antibiotic resistance paternity was also performed for strains of *Streptococcus* isolates.

Samples were incubated in BHI broth (Oxoid, Hampshire, UK), on Columbia agar with the addition of 5% defibrinated sheep blood (Oxoid, Hampshire, UK) and BD Chocolate Agar Columbia (Becton Dickinson, Heidelberg, Germany) under aerobic/microaerophilic (5% CO_2_) conditions at 37 °C for 24–48 h or sometimes up to 72 h.

The isolates with alpha-hemolytic colonies were further identified by conventional biochemical tests and API 20 Strep (bioMerieux SA, Marcy l’Etoile, France). Subsequently, the colonies were confirmed to be *Streptococcus* spp. by the matrix-assisted laser desorption-ionization-time of flight mass spectrometry (MALDI-TOF MS).

Before susceptibility testing, presumptive *Streptococcus* spp. strains were identified by matrix-assisted laser desorption-ionization-time of flight mass spectrometry (MALDI-TOF MS, Bruker Daltonik, Bremen, Germany).

#### MALDI-TOF MS Bacterial Identification

The ethanol/formic acid protocol was used to prepare MALDI-TOF MS samples. The bacterial protein suspension had a volume of 1 μL and was transferred to a MALDI target plate (Bruker, Daltonik, Germany). Then, 1 μL of matrix (10 mg α cyano-4-hydroxycinnamic acid ml-1 in 50% acetonitrile 2.5% trifluoroacetic acid) was added to the bacterial suspension. Bacterial mass spectra were acquired using the Microflex™ mass spectrometer (Bruker, Daltonik). For MALDI-TOF MS identification, the spectra captured were loaded into MALDI BioTyper™ 3.0 and compared to the manufacturer’s library. Standard Bruker interpretation criteria were applied. Scores ≥ 2.0 were accepted for species assignment, and scores ≥ 1.7 but ≤ 2.0 were used for genus identification [43].

### 4.3. Antimicrobial Susceptibility Testing

Agar disk diffusion was used to perform susceptibility testing on a total of 267 isolates, following the recommendations given by the CLSI documents M100 (Clinical and Laboratory Standards Institute, 2023) [71]. Antimicrobial substances were selected based on their interpretative criteria for certain species of *Streptococcus* spp., as well as their importance for both porcine and human health.

Each strain was subcultured on Columbia blood agar (5% sheep blood) and incubated overnight at a temperature of 37 °C with 5% CO_2_. Antimicrobial susceptibility was tested using the disc diffusion method on a Mueller–Hinton plate containing 5% sheep blood and according to the standards of Clinical Laboratory Standards Institute (CLSI) (VET01-A and VET01-S) [71,72]. To prepare an inoculum for each strain, an initial concentration of 1 × 10^8^ CFU/mL was used to inoculate the Mueller–Hinton plate before the application of antimicrobial discs (BIO-RAD, Marnes-La-Coquette, France). The plates were then incubated for 24 h at 35 °C. After bacterial growth, the inhibition zone diameters were measured.

The reference strain *Streptococcus pneumoniae* ATCC 49619 was used as a quality control according to CLSI recommendations.

The following antibiotics were used: ß-lactamine (penicillin, ampicillin, amoxicillin); β-lactamine with inhibitors (amoxicillin-clavulanic acid); cephalosporins (ceftiofur, cefquinome); polymyxin (colistin); aminoglycoside (apramycin, kanamycin, gentamicin, neomycin, streptomycin); tetracyclines (chlortetracycline, doxycycline); potentiated sulfonamides (trimethoprim sulfamethoxazole); fluoroquinolones (enrofloxacin, flumequine); amphenicols (florfenicol); macrolides (erythromycin, tilmicosin, tylosin, tulathromycin); lincosamide (lincomycin); lincomycin–spectinomycin and pleuromutiline (tiamulin).

Multidrug resistance (MDR) is defined as resistance to at least one agent from three or more antimicrobial categories, as classified by Magiorakos et al. [73].

## 5. Conclusions

To our knowledge, this is the first study in Romania on the isolation of streptococcal species other than *S. suis* from different pig pathologies.

Our results highlight the presence of atypical *Streptococcus* spp. in pig herds, which may represent a high and underestimated health risk for pigs but also for humans and should be correctly identified by more complex and sensitive veterinary diagnostic methods (MALDI-TOF MS).

We believe that further large-scale studies should investigate the impact of *SDSE* and *S. porcinus* on the pathogenesis of reproductive disorders in sows, in particular.

Nearly half of the isolated strains showed resistance to four or more groups of antibiotics.

The use of antibiotics in animals resulted in a selection of resistant bacteria in animals. To what extent antibiotic use and resistance in animals also influence the occurrence of resistance in humans remains an important area of debate. It is desirable that the use of antibiotics continues to decrease, and eventually for it to become a rare occurrence in pig farming. However, this will require additional efforts and a focus on better husbandry, biosecurity, and management. Ultimately, this reduction will lead to a decrease in resistance selection, which will benefit both animal and human health, global food safety, and food security.

## Figures and Tables

**Figure 1 antibiotics-13-00277-f001:**
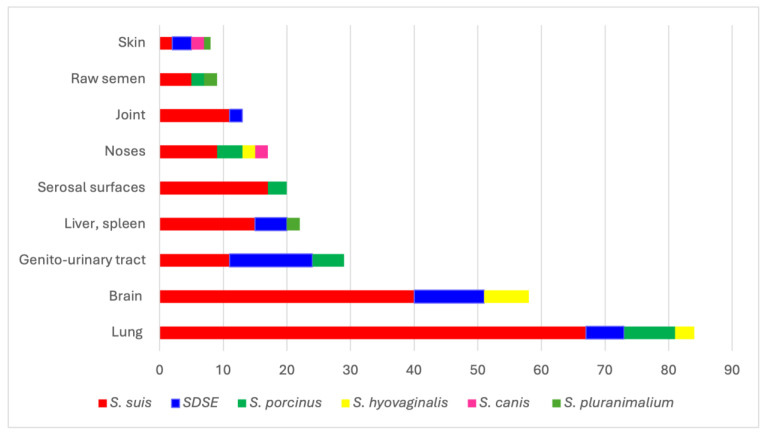
Distribution of the *Streptococcus* spp. isolated from various organs of pigs.

**Table 1 antibiotics-13-00277-t001:** Total number of isolated *Streptococcus* spp. from different sources.

No. crt.	Type of Sample/Source	Isolates No (%)	*Streptococcus Species*					
			*S. suis*	*S. dysgalactiae subsp*. *equisimillis*	*S. porcinus*	*S. hyovaginalis*	*S. pluranimalium*	*S. canis*
1.	Brain (CNS)	58 (21.72%)	40 (22.09%)	11 (27.50%)	0.00	7 (53.84%)	0.00	0.00
2.	Lung	84 (31.46%)	67 (37.01%)	6 (15.00%)	8 (33.33%)	3 (23.07%)	0.00	0.00
3.	Genitourinary tract	29 (10.86%)	11 (6.07%)	13 (32.50%)	5 (20.83%)	0.00	0.00	0.00
4.	Serosal surfaces	20 (7.49%)	17 (9.39%)	0.00	3 (12.50%)	0.00	0.00	0.00
5.	Noses	17 (6.36%)	9 (4.97%)	0.00	4 (16.66%)	2 (15.38%)	0.00	2 (50.00%)
6.	Skin	8 (2.99%)	2 (1.10%)	3 (7.50%)	0.00	0.00	1 (20.00%)	2 (50.00%)
7.	Joint	13 (4.86%)	11 (6.07%)	2 (5.00%)	0.00	0.00	0.00	0.00
8.	Liver, spleen	22 (8.23%)	15 (8.28%)	5 (12.50%)	0.00	0.00	2 (40.00%)	0.00
9.	Raw semen	9 (3.37%)	5 (2.76%)	0.00	2 (8.33%)	0.00	2 (40.00%)	0.00
10.	Other (heart, kidney)	7 (2.62%)	4 (2.20%)	0.00	2 (8.33%)	1 (7.69%)	0.00	0.00
	Total	267 (100.00%)	181 (67.79%)	40 (14.98%)	24 (8.98%)	13 (4.86%)	5 (1.87%)	4 (1.49%)

**Table 2 antibiotics-13-00277-t002:** Antimicrobial sensitivity of isolated *S. suis* strains.

Type of ATB	Disk Content	Tested Strains	Susceptible	Resistant	Intermediate
			% (No.)	% (No.)	% (No.)
Amoxicillin	AML (10 µg)	88	55.68 (49)	20.45 (18)	23.86 (21)
Amoxiclav	AMC (30 µg)	21	80.95 (17)	0.00	19.04 (4)
Ampicillin	AMP (10 µg)	22	72.72 (16)	13.63 (3)	13.63 (3)
Apramycin	APR (10 µg)	21	9.52 (2)	66.66 (14)	23.80 (5)
Ceftiofur	EFT (30 µg)	91	87.91(80)	3.29 (3)	8.79 (8)
Cefquinome	CEQ (30 µg)	28	82.14 (23)	7.14 (2)	13.04 (3)
Colistine	CT (10 µg)	77	2.59 (2)	53.24 (41)	44.15 (34)
Chlortetracycline	CTC (10 µg)	49	2.04 (1)	83.67 (41)	8.51 (7)
Doxycycline	DO (30 µg)	47	4.25 (2)	87.23 (41)	8.51 (4)
Enrofloxacin	ENR (5 µg)	95	47.36 (45)	21.05 (20)	31.57 (30)
Erythromycin	ERY (30 µg)	36	16.66 (6)	72.22 (26)	11.11 (4)
Florfenicol	FFC (30 µg)	86	53.48 (46)	31.39 (27)	15.11 (13)
Flumequine	UBN (30 µg)	12	0.00	50.00 (6)	50.00 (6)
Gentamycin	CN (10 µg)	49	26.53 (13)	26.53 (13)	46.93 (23)
Kanamycin	K (10 µg)	13	15.38 (2)	84.61 (11)	0.00
Lincomycin	LCN (15 µg)	20	0.00	95.00 (19)	5.00 (1)
Lincospectin	LS (10 µg)	10	80.00 (8)	0.00	20.00 (2)
Neomycin	N (10 µg)	86	10.46 (9)	48.83 (42)	40.69 (35)
Penicillin	P (10 UNITS)	68	66.17 (45)	16.17 (11)	17.64 (12)
Spectinomycin	SPT (100 µg)	18	0.00	94.44 (17)	5.55 (1)
Streptomycin	STRP (30 µg)	19	0.00	84.21 (16)	15.78 (3)
Tiamulin	TIAMU (30 µg)	11	36.36 (4)	45.45 (5)	18.18 (2)
Tilmicosin	TIL (15 µg)	7	0.00	85.71 (6)	14.28 (1)
Trimethoprim sulfamethoxazole	SXT (10 µg)	43	25.58 (11)	65.11 (28)	9.30 (4)
Tulathromycin	TUL (30 µg)	11	36.36 (4)	45.45 (5)	18.18 (2)
Tylosin	TYL (30 µg)	16	0.00	87.5 (14)	12.50 (2)

**Table 3 antibiotics-13-00277-t003:** The antimicrobial sensitivity patterns of the strains isolated from pigs.

Profiles	Antimicrobial Group with Resistance	No of Strains
1.	ß—lactams, Macrolides	18
2.	ß-lactams, Tetracyclines	9
3.	Tetracycline, Lincosamides	13
4.	ß-lactams, Tetracyclines, Fluoroquinolones	7
5.	Aminoglycosides, Macrolides, Sulphonamides	15
6.	ß-lactams, Tetracyclines, Fluoroquinolones, Macrolides	15
7.	Aminoglycosides, Macrolides, Tetracyclines, Lincosamides	9
8.	Aminoglycosides, Macrolides, Amphenicols, Sulphonamides	5
9.	Macrolides, Tetracyclines, Lincosamides, Sulphonamides	18
10.	Macrolides, Tetracyclines, Lincosamides, Aminoglycosides	6
11.	Macrolides, Tetracyclines, ß-lactams, Fluoroquinolones	21
12.	Aminoglycosides, Macrolides, Tetracyclines, Lincosamides, Aminocyclitol	8
13.	Macrolides, Aminoglycosides, Tetracyclines, ß-lactams with inhibitors, Pleuromutilins	42
14.	Macrolides, Aminoglycosides, Tetracyclines, Fluoroquinolones, Cephalosporins	33
15.	Macrolides, Aminoglycosides, Tetracyclines, Fluoroquinolones, Sulphonamides	8
16.	Macrolides, Aminoglycosides, Tetracyclines, Amphenicols, Sulphonamides, Pleuromutilins	22
17.	Macrolides, Aminoglycosides, Amphenicols, Sulphonamides, Fluoroquinolones, Lincosamides	18

**Table 4 antibiotics-13-00277-t004:** Number of *Streptococcus* spp. strains isolated from each age group.

Age Group		Suckling Piglets	Weaned Piglets	Fattening Piglets	Gilts/Sows	Boars	Total
	Sources						
Lung		15	49	16	2	2	84
CNS		11	39	4	3	1	58
Genitourinary tract		0	0	0	29	0	29
Liver, spleen		1	10	9	2	0	22
Serosal surfaces		0	12	5	3	0	20
Noses		0	11	5	1	0	17
Joints		2	9	2	0	0	13
Raw semen		0	0	0	0	9	9
Skin		0	4	3	1	0	8
Other (heart, kidney)		0	4	3	0	0	7
Total		29	138	47	41	12	267

## Data Availability

Data are contained within the article.

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
