# Peer review of "Multiple Drug Resistant Streptococcus Strains—An Actual Problem in Pig Farms in Western Romania"

_antibiotics, 2024, doi:10.3390/antibiotics13030277_

Round 1

Reviewer 1 Report

Comments and Suggestions for Authors

The article is interesting, however they need to separate the strain information and relate it to bacterial resistance and make some type of correlation of their results with the technology of the sampled animals (origin of the animal/type of production).

They discuss a lot but they have not related their samples to the origin of the animals. You cannot discuss the importance of the resistances found if you do not know the conditions in which the sampled pigs were, age, sex, severity, etc.

Your information is ambiguous if it cannot be correlated epidemiologically or clinically with animals.

All this information on the origin of the samples was not classified or correlated with anything in the study, there are only many samples evaluated with no epidemiological or clinical relevance.

Introduction

There is a lack of epidemiological information. It is not mentioned how serious or important they are? What is their incidence in the species mentioned? Their pathogenicity? The introduction is not very informative about their real importance. What is its incidence?

Conclusions

It is not mentioned what type of production the samples belonged to, a sample from a technical ranch is not the same as a backyard animal, their samples are not specific enough to be able to conclude and discuss everything they have mentioned, they must be a little more specific and separate the samples to be able to give a focus to everything mentioned and above all to be able to talk about resistance and all the possibilities of spreading infections as they discuss, a backyard pig is not the same as this one. in contact with other animals and people to a technical ranch in which the possibility of spread to other species, including man, decreases.

Comments on the Quality of English Language

Reviewer 2 Report

Comments and Suggestions for Authors

Dear Authors

Manuscript is inreresting but needs improvement.

Table 1: Could you change title ,, Total number of isolated Streptococcus spp. from different sources not organs because raw semen is not organ.

Results: line 137: Colistin is only actice against Gram negattive bacteria. So why authors tested for Gram positive bacteria?

Table 2 Why authors not tested all strain on all antibiotics? What was the reason?.

Disccusion

line 218: Could you explain for readers what does mean PRRS, PCV2.

Comments on the Quality of English Language

needs improvement.

Reviewer 3 Report

Comments and Suggestions for Authors

Dear Authors,

I am impressed from the enormous work you have done over the 5 year period. Your research is really important from the point of view of increasing microbial resistance to clinically administered antibiotics in pig farms for prevention and treatment. But I have a few comments and suggestions for you!

First, what is the purpose of the research? You didn't add it to the Introduction!

Second, you start with directly with the results (section Results). The reader needs to know what you've been up to! It was hard for me to understand too!

Third, nowhere in the Results do you mention that you confirmed the identity of the isolates by spectrophotometric analysis (MALDI-TOF MS)! 

In some places in the Discussion you mention "gene transfer", "virulence factors" etc. Have you done gene expression? If not, why are you even mentioning them?

Reviewer 4 Report

Comments and Suggestions for Authors

In the manuscript, authors report data obtained on the antibiotic resistance study of strains of Spreptococcus spp. isolated over 5 years.

Data in the manuscript are of great interest, but the form needs to be improved. The English language also needs to be revised

Abstract

lane 17: "Western Romania" I think the problem is not only in Romania, but is global

lane 19: Please, write Streptococcus spp.  and not only S. spp.

Introduction

lane 36: the introduction is not the right section in which to write what the authors use as a method

Please add the aims of the manuscript

Results

Table 1: Please, divide the S. pluranimalium and S. canis percentage. I think this is important in order to better understand the part of the discussion where the percentage of S. canis in humans is reported, and in any case the two pathogens are addressed in separate parts

Figure 1: Please, arrange the organs in increasing percentage order to make the results more intuitive

lane 111: "Among diseased pigs exhibiting nervous manifestations, S. suis was detected in 22.09% 111 of samples, followed by S. hyovaginalis at 53.84% and SDSE at 27.50% from the CNS." Is there a mistake? The percentages do not coincide with those shown in the table, and in any case, possibly, 53.84% is followed by 27.5% and 22.09%. Please, clarify

lane 124: "Other Streptococcus spp. were isolated from various sources" Which others?

lane 125: "S. suis was the most isolated species and it could be found in all sample types" This is not a result, please clarify or delete the second part of the sentence

lane 132: Please, insert the number of the table. In addition, it specifies that only the data of isolated S. suis loops were included in the table

Table 2: Please, Adjust title to 'Antimicrobial sensitivity of isolated S. suis strains'

Please format the table as the journal required

Discussion

I find the discussion section too long, so I recommend making it more concise.

lane 219: " These infections resulted in bacterial overgrowth" I think it's better to write " These infections could be resulted in bacterial overgrowth"

lane 261:  "isolated" and not "recover"

lane 281:Why do you think the liver could also be contaminated?

Lanes 316-325: I recommend deleting this part of the discussion, since in this manuscript only the phenotypic aspect of the isolated strains is discussed in detail

Materials and methods

I suggest the authors divide this section into: sampling, microbiological analysis, bacterial identification and antimicrobial susceptibility testing

lane 390: " were clinically affected" take the concept back to line 392. Please choose where to write it and don't be repetitive

lanes 403-406: these not are material and methods

lanes 413-417: Repeated concept

lane 421: Please, add a reference

Author Response

Please see gthe attachment

Round 2

Reviewer 2 Report

Comments and Suggestions for Authors

Dear Authors

Thank you for your revision.

Author Response

Dear Reviewer 2,

Thank you very much for appreciating our manuscript and for you kind observations.

Reviewer 3 Report

Comments and Suggestions for Authors

Dear Authors,

Thank you for your answers! As I mentioned before, you have done a great job! I think the research is important to track the increasing antimicrobial resistance in pig farms around the world! Because this problem occurs not only in Romania, but also in many other countries where the use of antibiotics is considered as prevention! I advise you to continue your research in this direction, focusing a little on gene expression to study the mechanism of transmission cycle animal-human-environment!

Author Response

Dear Reviewer 3,

Thank you very much for your pertinent and constructive comments. We will follow your advice further, indeed it is extremely important to know the elements that are included in the animal-human-environment triad.

Reviewer 4 Report

Comments and Suggestions for Authors

Authors responded adequately to all requests and improved the manuscript.

Despite this, in my opinion there is still some smallness to be fixed.

Table 4: Add the total of Streptococcus strains isolated at the end of the table and change the title. "Number of Streptococcus spp. strains that could be isolated from age group." please, change with "Number of Streptococcus spp. strains  isolated from different age groups"

lanes 394-396: please, move this sentence at the beginning of the next paragraph

Author Response

Dear Reviewer 4,

Thank you very much for your valuable and constructive comments. I have made the changes you suggested.